# Interhospital Spread of *bla*_VIM-1_- and *bla*_CTX-M-15_-Producing *K. pneumoniae* ST15 on an IncR Plasmid in Southern Spain

**DOI:** 10.3390/antibiotics12121727

**Published:** 2023-12-13

**Authors:** Patricia Pérez-Palacios, Ana Gual-de-Torrella, Ines Portillo-Calderón, Esther Recacha-Villamor, Francisco Franco-Álvarez de Luna, Lorena Lopez-Cerero, Alvaro Pascual

**Affiliations:** 1Division of Infectious Diseases and Microbiology, University Hospital Virgen Macarena, 41009 Seville, Spain; pperez5@us.es (P.P.-P.);; 2Institute of Biomedicine of Sevilla, 41013 Seville, Spain; 3Centro de Investigación Biomédica en Red de Enfermedades Infecciosas, Instituto de Salud Carlos III, 28029 Madrid, Spain; 4Servicio de Microbiología, Hospital Universitario Juan Ramón Jiménez, 21005 Huelva, Spain; 5Departamento de Microbiología, Facultad de Medicina, Universidad de Sevilla, 41009 Seville, Spain

**Keywords:** surveillance, carbapenemase spread, plasmids, *K. pneumoniae* high-risk clone

## Abstract

In 2014–2015, the main CTX-M-15- and OXA-48-producing clone in our region was ST15. Recently, *K. pneumoniae* ST15 isolates co-producing VIM-1 and CTX-M-15 were detected in several hospitals. The aim was to study the emergence and acquisition of this carbapenemase. Between 2017 and 2019, four hospitals submitted twenty-nine VIM-1- and CTX-M-15-producing *K. pneumoniae* ST15 isolates to our laboratory. Seven representatives of each *XbaI* PFGE pulsotype were sequenced using short- and long-read technologies. RAST, CGE databases, and Pathogenwatch were used for resistance determinants and capsule-type analysis. Plasmid comparison was performed with Easyfig2.1. Phylogenetic analysis included other contemporary ST15 isolates from Spain. The 29 isolates were clustered into seven different pulsotypes. The selected genomes, from three hospitals in two different provinces, were clustered together (fewer than 35 alleles) and differed by more than 100 alleles from other ST15 isolates obtained in the region. These seven isolates harbored one IncR plasmid (200–220 kb) with a common backbone and four regions flanked by IS*26*: one contained *bla*_VIM-1_, another contained *bla*_CTX-M-15_, the third contained *bla*_OXA-1_, and the fourth harbored heavy-metal-tolerance genes. The two initial plasmids, from two different centers, were identical, and rearrangement of four regions was observed in the five subsequent plasmids. Our findings showed the first intercenter dissemination of IncR plasmids carrying *bla*_VIM-1_, *bla*_CTX-M-15_, and metal-tolerance genes mediated by a new lineage of *K. pneumoniae* ST15. Two different capture events of the *bla*_VIM-1_ gene or different IS*26*-mediated plasmid rearrangements from a common ancestor may explain plasmid variations.

## 1. Introduction

Carbapenemase production is a serious health concern in infectious diseases and is becoming a real threat to global health [1]. An important part of this problem involves the increase in carbapenem resistance in Enterobacterales, which was declared a top priority for action by the WHO in 2017 [2]. Rapid identification and monitoring of carbapenem-resistant Gram-negative bacteria is necessary to tackle and control their spread [3]. Among Enterobacterales, *Klebsiella pneumoniae* is characterized by its extraordinary ability to spread in the community, spread in hospitals [4], and acquire carbapenemase genes. Dissemination has been associated in the last decade with broadly resistant clones with a high capacity for survival and niche colonization, as well as a considerable ability to capture virulence and resistance genes through mobile elements, such as plasmids and transposons, and to transfer them to their progeny or to other species [5].

The ST15 clone is one of the high-risk clones of *K. pneumoniae* that has worldwide distribution and, together with clones ST11, ST258/512, ST307, and ST147, is responsible for the spread of extended-spectrum beta-lactamases (ESBLs) and carbapenemases [6]. Associations between plasmids and prevalent bacterial clones are extremely frequent [7]. These particular associations have been extensively studied for *K. pneumoniae* ST11 and ST405 in association with the pOXA-48 plasmid [8], but similar studies for clone ST15 are scarce. In Andalusia, the ST15 and ST11 clones *of K. pneumoniae* have been one of the main lineages spreading *bla*_OXA-48_ and *bla*_CTX-M-15_ since 2014 [9]. Recently, isolates of this clone producing *bla*_VIM-1_, together with *bla*_CTX-M-15_, have been detected in several hospitals in our region. The aim of this study was to characterize these isolates, determine the phylogenetic relationships between them, and analyze the plasmid profile in order to determine whether the increase was due to several independent events or to the geographic spread of a single lineage.

## 2. Results

The 29 *K. pneumoniae* ST15 strains co-producing VIM-1 and CTX-M-15 were collected from different sources. Nineteen (65.5%) isolates were recovered from clinical samples: seven (24.1%) from urine samples, four (13.8%) from blood cultures, five (17.2%) from skin and soft-tissue samples, and three (10.3%) from respiratory samples. Ten (34.5%) isolates were collected from surveillance samples (rectal swabs) (Table 1).

By PFGE, seven pulsotypes were detected: P1 (*n* = 4), P2 (*n* = 3), P3 (*n* = 3), P4 (*n* = 7), P5 (*n* = 4), P6 (*n* = 1), and P7 (*n* = 7) (Appendix A). The first isolate, from a blood culture, was detected in August 2017 at the Hospital Infanta Elena in Huelva. One month later, a second isolate was detected in the same hospital in a urine sample. Both isolates were identical by PFGE. Six months later, in March 2018, a third isolate was detected in a rectal swab sample from a patient, transferred from Hospital Infanta Elena, at the Hospital Virgen Macarena in Seville (Figure 1). This third isolate was assigned to a different pulsotype. During 2018, seven more isolates were detected: six between May and November at the Hospital Infanta Elena, where all patients were admitted to the same medical unit, and the last one in December at the Hospital Virgen Macarena (the reference center for the other hospitals). In 2019, the number of isolates increased to 19, and these isolates were detected in four hospitals in our region: 3 isolates at Hospital Juan Ramón Jiménez (Huelva), 14 at the Hospital Macarena (Seville), 1 at the Hospital Infanta Elena (Huelva), and 1 at the Hospital de la Merced (Seville). The patients admitted to the Hospital Virgen Macarena had also been admitted to the same medical unit (Figure 1). As in the previous year, the origin of the isolates was diverse, and all were distributed among the seven different pulsotypes. The annual increase is shown in Table 1. All isolates were assigned to the KL64 O2v1 capsule type (Table 1). The following control measures were implemented in the four centers: terminal cleaning after patient discharge, isolation of positive patients, and electronic tagging of colonized patients for isolation in case of new admissions.

Despite the differences observed by PFGE, when the Ridom cgMLST scheme was used, the seven selected representative pulsotypes belonged to the same core-genome sequence (cgST3266) and were detected in three centers (Table 1). A close genetic relationship was found between the seven genomes: fewer than 10 alleles among six genomes recovered from two centers in the same province (Huelva) and the reference hospital (Hospital Virgen Macarena, Seville) and one isolate differed by 27–33 alleles (Appendix A). The latter isolate was recovered from a patient admitted to the reference hospital, and an epidemiological link was established with the Hospital Infanta Elena. All seven genomes were different (more than 100 alleles of difference) from other ST15 isolates from other centers in the region, as well from the two other VIM-1 producers detected in Spain in other regions in 2013–2014, indicating a new lineage.

All isolates were resistant to piperacillin/tazobactam, cefotaxime, cefepime, ertapenem, meropenem, imipenem, gentamicin, tobramycin, ciprofloxacin, and sulfamethoxazole-trimethoprim and susceptible to amikacin, fosfomycin, and colistin. In addition to the *bla*_CTX-M-15_
*and bla*_VIM-1_ genes, the isolates also harbored other determinants of resistance: to aminoglycosides (*aac(3)-IIa*, *aac(6*′*)-Ib-cr, aph(6)-Id, aph(3*″*)-Ib,* and *aph(3*′*)-Ia*), quinolones (*qnrB1* and *qnrB2*), and folate antagonists (*sul1*, *sul2,* and *dfrA12*) (Appendix A). 

Plasmid analysis of the seven genomes revealed that the *bla*_VIM-1_ and *bla*_CTX-M-15_ genes were on the same type of plasmid belonging to the IncR incompatibility group and approximately 200,000 pb in size (Figure 2). Comparative genomic analysis showed that the seven plasmids shared the same backbone and three identical regions flanked by IS*26*s. The two plasmids from 2017 and 2018 were identical but came from two different centers in Huelva. In the rest of the five plasmids, the order of the four regions within the plasmids was different and showed rearrangements (Figure 2). The *bla*_VIM-1_ region (approx. 20,000 bp) was located in a class 1 integron, flanked on each side by IS*26*. Next to *bla*_VIM-1_, other antimicrobial-resistance genes, to aminoglycosides (*aac(6*′*)-Ib-cr* and *aadA1*), sulfonamides (*sul1*), and quinolones (*qnrB2*), were detected (Figure 2). The regions containing *bla*_CTX-M-15_ (approx. 28,000 bp) and *bla*_OXA-1_ (approx. 12,000 bp) were also flanked on each side by IS*26*. The *bla*_CTX-M-15_ region included *bla*_TEM-1B,_ and other aminoglycoside-resistance genes (*aph(3*′*)-Ib*, *aph(6*′*)-Id*, and aac(*3*′)-Ib-cr) and the *bla*_OXA-1_ region included the *aac(6*′*)-Ib-cr* gene. In addition, a third region of approx. 40,000 bp, also flanked by IS*26*, carried the copper-resistance operon *(copABCDRSE)*, silver-tolerance (*silESRCFBAP*) genes, and the arsenic-resistance operon (*arsRDABCH*) (Figure 2).

## 3. Discussion

In this study, the characteristics of *K. pneumoniae* ST15 co-producing *bla*_VIM-1_ and *bla*_CTX-M-15_ on an IncR plasmid indicate that this is a new lineage within clone ST15 that has been spreading among different hospitals in our region since 2017. This clone was found prevalent in our region during 2014–2015 but was mainly associated with the resistant determinants *bla*_CTX-M-15_ and *bla*_OXA-48_ [9].

With respect to the specific association between clones and plasmids, about which we have less knowledge in ST15, IncR has been linked to the dissemination of the clone ST15 and CTX-M-15 in Portugal (six isolates) and in four geographical areas of Bulgaria [10,11]. In this study, all isolates carried an IncR plasmid, which, in addition to harboring the *bla*_VIM-1_- and *bla*_CTX-M-15_-resistance determinants*,* also carried others encoding resistance to aminoglycosides, sulfonamides, and quinolones. The same plasmid was also found to be prevalent among ST15 isolates in our area in 2014–2015 [9]. These findings suggest that the current VIM-1- and CTX-M-15-producing isolates may have emerged from a population already endemic in our area that carried CTX-M-15. On the other hand, several genes for resistance to copper, silver, and arsenic were also detected in this plasmid. The presence of metal and antibiotic-resistance genes in the same mobile elements could imply contaminated environmental reservoirs exerting positive selection pressure on tolerant bacteria, favoring the co-transfer of carbapenemase and metal genes [12].

In our region, since 2014 and 2015, *K. pneumonia* ST15 has mainly co-produced *bla*_OXA-48,_ located on IncM and IncL plasmids, and *bla*_CTX-M-15,_ located on IncR plasmids [9]. The presence of *bla*_VIM-1_ in a region flanked by IS*26* could indicate that the IncR plasmid containing *bla*_CTX-M-15_ captured this carbapenemase, which resulted, over time, in different rearrangements within the plasmid. IS*26* is a mobile element of great importance in the dissemination of antibiotic-resistance determinants in *Enterobacterales*, driving rearrangements and remodeling within the plasmid [13]. Replication of the *IS*26 transposase, Tnp*26*, results in the movement of sites flanked by this element, leading to mobilization to new sites and deletion or inversion of adjacent DNA [14]. The dynamic evolution of plasmids and regions mediated by IS*26* has already been described; it has also been suggested that *bla*_SHV_, originally on the *K. pneumoniae* chromosome, was mobilized to a plasmid by an IS*26*-mediated event [15,16]. Further studies are needed to understand the implications of these genetic events for the evolution and diversity of plasmids containing carbapenem-resistance genes when IS*26* is present [17].

Studies describing outbreaks caused by *K. pneumoniae* ST15 have mainly been associated with *bla*_KPC-2_- and *bla*_VIM-1_-type carbapenemases and have focused on studying its capacity for dissemination in intensive-care-unit patients and on the great difficulty of controlling its spread [18,19,20]. There has also been one study reporting inter-regional spread of *K. pneumoniae* ST15, although in that case, it was only associated with ESBL and was the result of transferring patients between different hospitals [21]. To date, intercenter dissemination of *bla*_VIM-1_- and *bla*_CTX-M-15_-producing isolates has not been described. VIM-1-producing *K. pneumoniae* ST15 has been detected previously in our country, in a European study conducted in 2013–2014, but those isolates were different from the ones found in this study. In our case, the close genetic relatedness of six of the seven isolates could indicate transmission due to regional spread, although data from an epidemiological survey of all cases are not available. Transmission could have occurred during routine transfers between centers: all the isolates were recovered either from two centers in the same city or from the reference hospital. Although the close relationship is suggestive of intrahospital transmission, genetic proximity could also be due to admissions of already colonized patients from other hospitals with which there is a high frequency of patient exchanges, as was observed in the genetically nearest-neighbor analysis carried out by David et al. [22]. 

Controlling the spread of multidrug-resistant bacteria is currently one of the challenges facing global epidemiology. Therefore, measures and interventions aimed at preventing interregional transmission are very important [23]. One measure that has been shown to have an impact on the control of multidrug-resistant microorganisms is the screening of patients on admission, especially in critical services such as intensive care [24]. Since 2018, colonized patients are routinely electronically flagged in our local region: any colonized patient is preventively isolated on every new admission to any center in the region, and the alert is maintained for 12 months. Measures of this type could have been the first line of defense to prevent the spread of certain successful clones, although full implementation was delayed in the early years, hindering rapid detection among new admissions and enabling the intercenter spread observed in 2019. 

A major limitation of this study is that isolates are submitted to the reference laboratory on a voluntary basis and, although the Regional Ministry of Health recommends that all carbapenemase-producing isolates be tested, it is not mandatory. Another important limitation of our study is the lack of epidemiological data on the cases, so there is a possibility of discrepancies between the results obtained in the genetic studies and the epidemiological data. Therefore, in order to have an overall view of the problem of intercenter dissemination, every effort should be made to collect both genetic and epidemiological data. Moreover, the carrier status of the patients was not known prior to admission, so it was difficult to determine the time of acquisition and direction of spread between centers.

## 4. Materials and Methods

### 4.1. Bacterial Strains: Collection and Identification

Between 2017 and 2019, four hospitals submitted a total of 29 isolates of VIM-1- and CTX-M-1-producing *K. pneumoniae* ST15 to the Regional Reference Laboratory for the Surveillance and Control of Nosocomial Infections and Prudent Use of Antimicrobials (PIRASOA) program (Hospital Universitario Virgen Macarena, Seville) for the region of Andalusia, Spain. The isolates originated from 4 hospitals located in 2 adjacent provinces (Seville and Huelva) in Western Andalusia: Hospital Virgen Macarena (*n* = 16) and Hospital de la Merced (*n* = 1) in Seville and Hospital Juan Ramón Jiménez (*n* = 3) and Hospital Infanta Elena (*n* = 9) in Huelva (Table 1). Hospital Virgen Macarena is the reference hospital for several surgeries (cardiac, thoracic, dermatologic, maxillofacial, plastic, and ophthalmic) for the other three hospitals. All isolates were identified by using MALDI-TOF MS (MALDI-TOF Biotyper 3.1; Microflex Bruker, Madrid, Spain).

### 4.2. Antimicrobial Susceptibility Testing

Antimicrobial susceptibility testing of all isolates was performed using the commercial MicroScan NMDRM1 panel (Beckman Coulter, West Sacramento, CA, USA). Assignment to a clinical category was based on EUCASTv21 breakpoints. Preliminary detection of carbapenemase production was carried out by enzymatic assay (β-carba, Biorad, Marnes-la-Coquette—France) and lateral immunochromatography (NG-Test CARBA-5, NG Biotech, Guipry, France).

### 4.3. Molecular Typing

Clonal relatedness was initially performed by pulsed-field-gel-electrophoresis (PFGE) analysis of *XbaI*-digested DNA (http://www.cdc.gov/pulsenet accesed on 10 November 2023). Isolates differing by one or more bands by PFGE were assigned to different pulsotypes [25]. A dendrogram was created using BioNumerics v.7.6 software (Applied Maths, Marcy-l’Étoile, France) using the Dice coefficient with 0.8% optimization and 1% band-position tolerance (Appendix A). A representative isolate from each pulsotype was selected for short- and long-read whole-genome sequencing; 3 of the 4 centres were covered.

Ridom SeqSphere+ (v8.1.0) was used for core-genome (cgMLST) and whole-genome (wgMLST) genotyping, as well as minimum-spanning-tree construction, comparing 2358 loci and considering sequence types that differed by more than 15 alleles as different. For comparison purposes, 13 contemporary genomes belonging to ST15 from the Andalusian Reference Regional Laboratory and other centers in the region were added (9 CTX-M-15-producers, 2 hyperproducers of SHV-1, and 2 OXA-48 and CTX-M-15 co-producers), as well as 2 genomes of VIM-1-producing ST15 isolates from other Spanish regions included in a European survey conducted in 2013 and 2014 [26]. A cut-off threshold of 10 alleles in the wgMLST Ridom scheme was used to define an interhospital transmission event [27,28]. A matrix of allele differences was constructed using the whole genome and the first VIM-1- and CTX-M-15-producing *K. pneumoniae* ST15 isolate as a reference (Appendix A).

### 4.4. Whole-Genome Sequencing (WGS)

DNA extraction was performed with the MagCore HF16 Plus automatic system (RCB Bioscience^®^, Nottingham, UK). For short-read sequencing, library preparation was performed using the Nextera Flex DNA library preparation kit (Illumina^®^, San Diego, CA, USA). DNA sequencing was performed with the MiSeq Reagent Kit V3 (Illumina^®^, San Diego, CA, USA) (600 cycles) and the Illumina MiSeq sequencer (2 × 300 paired-end reads). Raw reads were trimmed, quality filtered, and then assembled de novo with CLC Genomic Workbench v9 (Qiagen, Beverly, MA, USA).

Annotation of antimicrobial-resistance determinants was carried out using ResFinder 4.1 (https://cge.cbs.dtu.dk/services/ResFinder/ (accessed on 14 November 2023)) (Appendix A). ST was determined using MLSTFinder databases (https://cge.cbs.dtu.dk/services/MLST/ (accessed on 14 November 2023)) [29] and capsule loci with Pathogenwatch [30].

### 4.5. Plasmid Analysis

The plasmid content of the isolates provided by short reads was analyzed using PlasmidFinder 2.1 (https://cge.cbs.dtu.dk (accessed on 14 November 2023)) and multilocus sequence typing with pMLST (Table 1). For complete plasmid analysis, a long-read approach was used. Each representative pulsotype was sequenced using single-molecule real-time (SMRT) technology (PacBio, Sequel II system, San Diego, CA, USA) and Flye (https://usegalaxy.org/ (accessed on 14 November 2023)) for de novo assembly. Further plasmid annotation was performed using the Bakta annotation pipeline [31] and further polished with the Basic Local Alignment Search Tool (BLAST) and *IS*finder. All sequences were deposited in the ENA database (Study ID: PRJEB52486). Plasmid sequences were compared and visualized using Easyfig2.1 [32].

## 5. Conclusions

Our data indicate the emergence and dissemination of a new lineage of ST15 *K. pneumoniae* co-producing *bla*_VIM-1_ and *bla*_CTX-M-15_ in Andalusian hospitals. The acquisition of a new resistance determinant in an area in which CTX-M-15-producing ST15 is endemic may facilitate spread between centers.

## Figures and Tables

**Figure 1 antibiotics-12-01727-f001:**
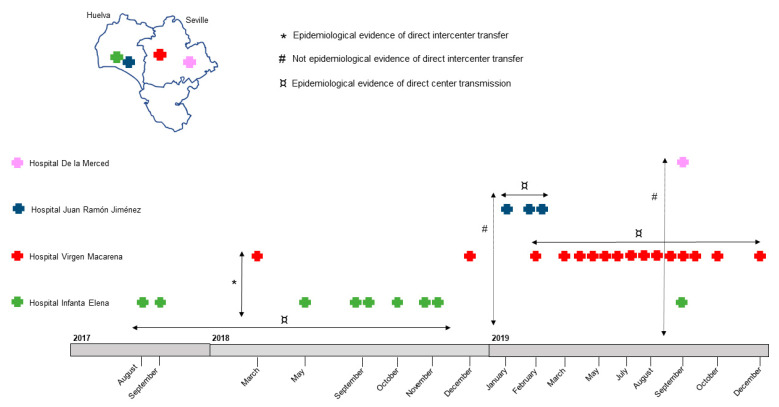
Graphical representation of possible pathways of interhospital spread for this study.

**Figure 2 antibiotics-12-01727-f002:**
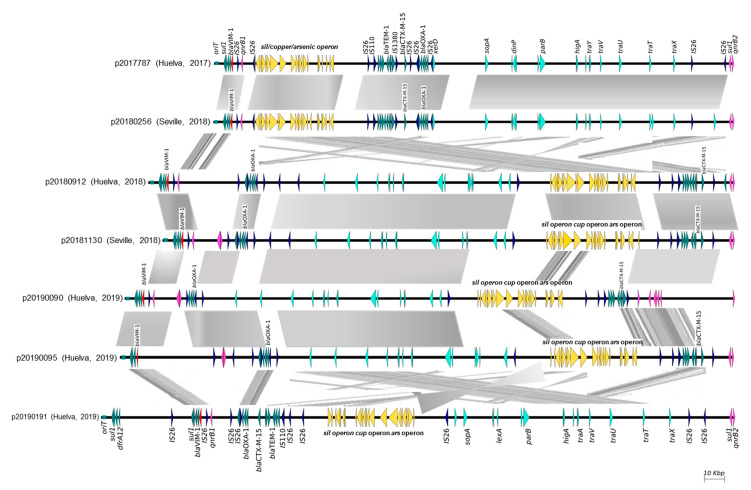
IncR plasmids harboring *bla*_VIM-1_ and *bla*_CTX-M-15_ from 7 *K. pneumoniae* ST15 isolates. *bla*_VIM-1_ is shown in red, *bla*_CTX-M-15_, *bla*_OXA-1_ and *bla*_TEM-1_ are shown in dark green, IS are shown in blue and metal resistance operons are shown in yellow.

**Table 1 antibiotics-12-01727-t001:** Relevant characteristics of the isolates of *K. pneumoniae* ST15 producing *bla*_VIM-1_ and *bla*_CTX-M-15._

Isolate (n°)	Isolation Date	Pulse Type	*bla*		Source	Hospital	cgMLST	Capsule Type	Plasmids
			ESBL ^1^	CPase ^2^					
2017787	14 August 2017	P1	CTX-M-15	VIM-1	Bloodstream	Infanta Elena	3266	KL64/O2v1	IncR, ColpVC
2017917	15 September 2017	P1	CTX-M-15	VIM-1	Urine	Infanta Elena	N.A ^3^		N.A
20180256	16 March 2018	P2	CTX-M-15	VIM-1	Rectal swab	Virgen Macarena	3266	KL64/O2v1	IncR, ColpVC
20180516	24 May 2018	P2	CTX-M-15	VIM-1	Bloodstream	Infanta Elena	N.A		N.A
20180784	5 September 2018	P3	CTX-M-15	VIM-1	Skin and soft tissues	Infanta Elena	N.A		N.A
20180797	15 September 2018	P3	CTX-M-15	VIM-1	Skin and soft tissues	Infanta Elena	N.A		N.A
20180912	5 October 2018	P3	CTX-M-15	VIM-1	Rectal swab	Infanta Elena	3266	KL64/O2v1	IncR, ColpVC
20181086	29 November 2018	P1	CTX-M-15	VIM-1	Urine	Infanta Elena	N.A		N.A
20181087	30 November 2018	P1	CTX-M-15	VIM-1	Rectal swab	Infanta Elena	N.A		N.A
20181130	13 December 2018	P4	CTX-M-15	VIM-1	Bloodstream	Virgen Macarena	3266	KL64/O2v1	IncR, ColpVC
20190090	31 January 2019	P5	CTX-M-15	VIM-1	Respiratory	Juan Ramón Jimenez	3266	KL64/O2v1	IncR, ColpVC
20190095	7 February 2019	P6	CTX-M-15	VIM-1	Urine	Juan Ramón Jimenez	3266	KL64/O2v1	IncR, ColpVC
20190096	13 February 2019	P7	CTX-M-15	VIM-1	Urine	Virgen Macarena	N.A ^3^		N.A
20190175	13 March 2019	P4	CTX-M-15	VIM-1	Urine	Virgen Macarena	N.A		N.A
20190191	22 February 2019	P7	CTX-M-15	VIM-1	Respiratory	Juan Ramón Jimenez	3266	KL64/O2v1	IncR, ColpVC
20190260	2 May 2019	P4	CTX-M-15	VIM-1	Rectal swab	Virgen Macarena	N.A		N.A
20190268	7 May 2019	P5	CTX-M-15	VIM-1	Rectal swab	Virgen Macarena	N.A		N.A
20190286	12 May 2019	P7	CTX-M-15	VIM-1	Bloodstream	Virgen Macarena	N.A		N.A
20190303	17 May 2019	P4	CTX-M-15	VIM-1	Rectal swab	Virgen Macarena	N.A		N.A
20190402	1 July 2019	P7	CTX-M-15	VIM-1	Rectal swab	Virgen Macarena	N.A		N.A
20190403	1 July 2019	P7	CTX-M-15	VIM-1	Rectal swab	Virgen Macarena	N.A		N.A
20190504	24 July 2019	P4	CTX-M-15	VIM-1	Skin and soft tissues	Virgen Macarena	N.A		N.A
20190523	14 August 2019	P4	CTX-M-15	VIM-1	Rectal swab	Virgen Macarena	N.A		N.A
20190524	14 August 2019	P4	CTX-M-15	VIM-1	Rectal swab	Virgen Macarena	N.A		N.A
20190526	14 August 2019	P7	CTX-M-15	VIM-1	Skin and soft tissues	Virgen Macarena	N.A ^3^		N.A
20190643	5 September 2019	P2	CTX-M-15	VIM-1	Respiratory	Infanta Elena	5114	KL64/O2v1	IncR, ColpVC
20190693	24 September 2019	P5	CTX-M-15	VIM-1	Skin and soft tissues	De la Merced	N.A		N.A
20190717	14 October 2019	P7	CTX-M-15	VIM-1	Urine	Virgen Macarena	N.A		N.A
20190944	20 December 2019	P5	CTX-M-15	VIM-1	Urine	Virgen Macarena	N.A		N.A

^1^ ESBL: extended-spectrum β-lactamase. ^2^ CPase: carbapenemases. ^3^ N.A: not analyzed.

## Data Availability

Data are contained within this article and Appendix A. WGS data are available in the ENA database (Bioproject ID: PRJEB52486).

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
