# Peer review of "Interhospital Spread of blaVIM-1- and blaCTX-M-15-Producing K. pneumoniae ST15 on an IncR Plasmid in Southern Spain"

_antibiotics, 2023, doi:10.3390/antibiotics12121727_

Round 1

Reviewer 1 Report

Comments and Suggestions for Authors

The manuscript describes genomic study of 29 K. pneumoniae ST15 isolates co-producing VIM-1 and CTX-M-15 from several Spanish hospitals demonstrating interhospital spread for the first time for this particular clone of K. pneumoniae. Practical significance of the study is high, since the results suggest implementation of specific measures for preventing further spread of multiresistant clones. The data reported are also novel, since similar studies were not published previously.

Methodology used to determine at first pulsotypes and then to make detailed sequencing of both bacterial chromosomes and plasmids is reliable and previously validated, so the results could be trusted. Remarkable genetic similarity between the isolates and obvious genetic difference from other isolates previously captured confirms the results and supports the conclusion about interhospital spread. 

The manuscript deserves acceptance, with only a few minor corrections:

1. the dendrogram mentioned in the Methods section was not presented in the supplementary files, so please add it

2. the Table1 was tripled, so please delete the redundant copies

3. the authors should add a scheme in the manuscript itself showing the most probable pathways of interhospital spread

4. main limitation of the study is lack of epidemiological information about the cases; please elaborate this more in the paragraph about limitations in the Discussion (including possibility of differences of results between genetic and epidemiological studies), and underline that authors of future studies should make every effort to collect both genetic and epidemiological data, since this would give more clear picture of the problem of interhospital spread.

Author Response

Please find attach our responds to your commentaries

Reviewer 2 Report

Comments and Suggestions for Authors

I appreciate the opportunity to review this manuscript.

Overall, the authors produce an interesting paper. The aim was to study the emergence and acquisition of carbapenemase. To do this, 29 isolates of VIM-1 and CTX-M-15-producing K. pneumoniae ST15 were first analysed by pulsed field electrophoretic analysis (PFGE) to determine their clonal relatedness. Subsequently, antimicrobial resistance determinants were detected by WGS and plasmids were analyzed by single molecule real-time (SMRT) technology.

Table 1: Table 1 should be a single table, not a split into 3 tables.

Lines 130-131: The authors distinguish clinical and surveillance isolates (see also Table 1). It would be helpful to clarify what they mean by surveillance isolates, from what kind of matrices were they isolated?

Figure 1: Please fix the figure. The plasmid identifiers are visible but the origin and date are covered by the image

Discussions lines 4-5: please rephrase, it is not clear what the authors mean

Author Response

(The authors gave the same response as above.)
